# Timosaponin A3 Induces Anti-Obesity and Anti-Diabetic Effects In Vitro and In Vivo

**DOI:** 10.3390/ijms25052914

**Published:** 2024-03-02

**Authors:** Ji-Hyuk Park, Wona Jee, So-Mi Park, Ye-Rin Park, Seok Woo Kim, Hanbit Bae, Won-Suk Chung, Jae-Heung Cho, Hyungsuk Kim, Mi-Yeon Song, Hyeung-Jin Jang

**Affiliations:** 1Department of Clinical Korean Medicine, Graduate School, Kyung Hee University, Seoul 02447, Republic of Korea; dr.jihyuk.park@gmail.com (J.-H.P.); omdluke@khu.ac.kr (W.-S.C.); vetkong95@hanmail.net (J.-H.C.); kim0874@hanmail.net (H.K.); 2Department of Korean Rehabilitation Medicine, College of Korean Medicine, Kyung Hee University, Seoul 02447, Republic of Korea; 3Department of Science in Korean Medicine, Graduate School, Kyung Hee University, Seoul 02447, Republic of Korea; 97wona@naver.com (W.J.); psm991030@naver.com (S.-M.P.); dutndi@naver.com (Y.-R.P.); kim66470@naver.com (S.W.K.); gksqlc4321@naver.com (H.B.); 4College of Korean Medicine, Kyung Hee University, 26, Kyungheedae-ro, Dongdaemun-gu, Seoul 02447, Republic of Korea

**Keywords:** anti-diabetic effect, anti-obesity effect, GLP-1, high-fat diet, Jimo, Timosaponin A3

## Abstract

Obesity is a serious global health challenge, closely associated with numerous chronic conditions including type 2 diabetes. *Anemarrhena asphodeloides* Bunge (AA) known as Jimo has been used to address conditions associated with pathogenic heat such as wasting-thirst in Korean Medicine. Timosaponin A3 (TA3), a natural compound extracted from AA, has demonstrated potential therapeutic effects in various disease models. However, its effects on diabetes and obesity remain largely unexplored. We investigated the anti-obesity and anti-diabetic properties of TA3 using in vitro and in vivo models. TA3 treatment in NCI-H716 cells stimulated the secretion of glucagon-like peptide 1 (GLP-1) through the activation of phosphorylation of protein kinase A catalytic subunit (PKAc) and 5′-AMP-activated protein kinase (AMPK). In 3T3-L1 adipocytes, TA3 effectively inhibited lipid accumulation by regulating adipogenesis and lipogenesis. In a high-fat diet (HFD)-induced mice model, TA3 administration significantly reduced body weight gain and food intake. Furthermore, TA3 improved glucose tolerance, lipid profiles, and mitigated hepatic steatosis in HFD-fed mice. Histological analysis revealed that TA3 reduced the size of white adipocytes and inhibited adipose tissue generation. Notably, TA3 downregulated the expression of lipogenic factor, including fatty-acid synthase (FAS) and sterol regulatory element-binding protein 1c (SREBP1c), emphasizing its potential as an anti-obesity agent. These findings revealed that TA3 may be efficiently used as a natural compound for tackling obesity, diabetes, and associated metabolic disorders, providing a novel approach for therapeutic intervention.

## 1. Introduction

Glucagon-like peptide 1 (GLP-1) is an incretin hormone composed of 30 or 31 amino acids. Discovered in 1983 as a derivative of proglucagon, GLP-1 is secreted by enteroendocrine L cells situated in the small intestine. GLP-1 plays a pivotal role in regulating glucose homeostasis and its many combined effects culminate in weight loss [1,2,3]. Numerous animal experiments and clinical trials have substantiated the effectiveness of stimulating GLP-1 secretion in the treatment and prevention of obesity [4,5,6,7]. Consequently, drugs that induce GLP-1 secretion have emerged as promising therapeutic agents for obesity. However, many studies have found that the use of those chemical drugs can cause disturbing symptoms and serious diseases as side effects [8,9,10,11]. This has led to a growing interest in the use and study of traditional herbal remedies as alternatives. In the context of obesity and diabetes treatment, there is a growing focus on naturally derived compounds that offer relative safety.

*Anemarrhena asphodeloides* Bunge (AA), known as Jimo in traditional Korean medicine and Zhimu in traditional Chinese medicine, is a perennial herb belonging to the lily family (Liliaceae) [12]. The dried rhizome of Jimo is employed as an herbal remedy in East Asia. Research has unveiled its pharmacological effects, which include blood glucose reduction and anti-pyretic properties [13,14,15]. The extract was reported to stimulate the secretion of insulin in diabetic rat models [16]. Also, there is a study which demonstrated that AA extract, particularly the ethyl acetate fraction, was effective as a stimulant of GLP-1 secretion in NCI-H716 cells by the alteration of gene expression. These results suggest the possibility of AA extracts as a therapeutic herbal medicine for type 2 diabetes treatment by regulating the secretion of GLP-1 [17].

In traditional Korean medicine, Jimo has been widely utilized as it presents anti-inflammatory, anti-pyretic, diuretic, and analgesic properties. Jimo features prominently in various traditional prescriptions such as Sanjoin-tan, Jaeumganghwa-tang, Baekho-tang, and others [18]. Examination of these prescriptions in Dongeuibogam, the most authoritative book in Korean medicine, reveals that Jimo is primarily employed to address conditions associated with pathogenic heat, which signifies an imbalance of heat within the body, and this imbalance can manifest in various forms such as a sensation of heat, fever, inflammation, infections, and skin issues. Moreover, Jimo is often combined with other medical herbs to treat an array of conditions, such as moistening the lungs to suppress coughing, nourishing Yin and bringing down fire, engendering fluid to suppress thirst, and clearing heat and purging fire, according to Dongeuibogam [19]. Classified as a cooling agent, Jimo is a key component in traditional remedies for feverish conditions. Considering that heat-related ailments are common, especially among So-yang type individuals with obesity or diabetes, as wasting-thirst according to Sasang constitutional medicine, Jimo has a valuable role in traditional Korean medicine [20].

Timosaponin A3 (TA3), a natural steroidal saponin isolated from Jimo, exhibits a diverse range of pharmacological activities. Research has elucidated its impact on various cellular signaling pathways and its efficacy in different cell types and disease models, including cancer, Alzheimer’s disease, depression, diabetes mellitus, and colitis [21]. It shows that TA3 is the most valuable compound pharmacologically which we can find from Jimo.

Nevertheless, our understanding of the anti-obesity effects and the underlying molecular mechanisms of TA3 remains limited. In this study, we aimed to address this knowledge gap by exploring the role of TA3 in obesity and diabetes, with potential implications for its clinical use as a medicine or as a therapeutic agent in the development of obesity and diabetes treatments.

## 2. Results

### 2.1. Timosaponin A3 Stimulates GLP-1 Secretion in NCI-H716

First, a cytotoxicity assessment at NCI-h716 was performed. Given the short drug-exposure duration (2 h), the toxicity was evaluated up to high concentrations. Ultimately, up to 20 μM of TA3 did not significantly affect cell viability (Figure 1A). To confirm the hypothesis that TA3 stimulates GLP-1 secretion in enteroendocrine L cells, TA3 was applied up to 10 μM, and the medium was collected to measure GLP-1 secretion using ELISA. A concentration-dependent increase in GLP-1 secretion was observed (Figure 1B). Subsequently, the protein expression of key signaling molecules in the GLP-1 secretion pathway was investigated. Increased phosphorylation of PKAc and AMPK was confirmed (Figure 1C). These results collectively suggest that TA3 does not exert significant cytotoxicity at concentrations up to 20 μM and significantly enhances GLP-1 secretion in a concentration-dependent manner in NCI-H716 cells. Furthermore, TA3 treatment leads to the increased phosphorylation of PKAc and AMPK, indicating its role in GLP-1 secretion.

### 2.2. Timosaponin A3 Suppresses Adipogenesis and Lipogenesis in 3T3-L1 Adipocytes

To assess cytotoxicity in 3T3-L1 adipocytes to TA3, a preliminary evaluation was conducted. Although the total drug exposure time was 8 days, toxicity was assessed after 48 h owing to media changes every 2 days. The results showed that no significant toxicity was observed up to 10 μM during the 48-h exposure. Thus, concentrations up to 10 μM were used in subsequent experiments (Figure 2A). Next, Oil Red O staining was performed to investigate the involvement of TA3 in adipogenic differentiation. The group treated with differentiation inducers showed a significant increase in lipid droplets and absorbance compared to the undifferentiated group. However, when treated with TA3, a concentration-dependent decrease in both lipid droplets and absorbance was observed. To further understand how TA3 inhibits adipogenesis in adipocytes, the expression of lipogenic factors, such as FAS and FABP4, was examined. These factors, which had increased owing to differentiation-inducer treatment, showed reduced expression levels upon TA3 treatment (Figure 2C). Key transcription factors involved in adipogenesis, C/EBPα and PPARγ, also exhibited increased expression following differentiation inducer treatment but were downregulated upon TA3 treatment (Figure 2D). Moreover, AMPK, a regulator of adipocyte differentiation and energy metabolism, showed dose-dependent increased expression upon TA3 treatment compared to the group treated with differentiation inducers alone (Figure 2E). In summary, the data collectively suggest that TA3 effectively inhibits adipogenesis, as evidenced by reduced lipid accumulation and the altered expression of key factors involved in adipocyte differentiation and lipogenesis.

### 2.3. Timosaponin A3 Attenuates Body Weight Gain and Food Intake 

To assess the in vivo anti-obesity effect of TA3, a high-fat diet (HFD) experiment was conducted. Following a 1-week adaptation period, the animals were randomly divided into three groups (n = 5 each): the ND group, the HFD group, and the HFD + TA3 10 mg/kg treatment (TA3) group. The animal experiment schedule was conducted as shown in Figure 3. Weekly body weight measurements revealed that body weight significantly increased in the HFD group compared to that in the ND group by the 8th week of the HFD. However, after 4 weeks of 10 mg/kg TA3 oral administration, body weight gradually decreased compared to the HFD group, showing a significant difference in the final 12th week (Figure 4A). Indeed, visual observations of the body images of each group taken before dissection also revealed noticeable differences (Figure 4B). While the food intake (g) appeared notably lower in the HFD group compared to that in the ND group, there was a reduction in food consumption in the TA3-treated group, although this was evident (Figure 4C). However, when converted to kcal and recalculated, it was evident that the HFD group consumed significantly more kcal compared to the ND group. Furthermore, the administration of TA3 resulted in a substantial reduction in kcal intake compared to the corresponding HFD group (Figure 4D). 

### 2.4. Improvement in Glucose Homeostasis and Lipid Profiles 

To evaluate the effect of TA3 on hyperlipidemia, we performed an OGTT. After a 16-h fasting period, the following day, 5 g/kg of glucose was orally administered, and the blood glucose levels were checked at 15, 30, 60, and 120 min. As a result, the HFD group maintained higher blood glucose concentrations for a longer period after the glucose intervention compared to the corresponding intervention in the ND group. In contrast, the TA3 group had a significant decrease in blood sugar concentration compared to that in the HFD group (Figure 5A). When comparing the AUC obtained to evaluate the OGTT, the HFD group showed higher measurements compared to the ND group, and the TA3 group exhibited a significant difference compared to the HFD group. (Figure 5B). All lipid profile factors were significantly elevated in the HFD-treated group compared to those in the ND group (Figure 5C–H). However, the concentrations of aspartate transferase (AST), alanine transaminase (ALT), TC, TG, and LDL were significantly reduced in the TA3 group compared to those in the HFD group. The difference between the concentration of HDL in the TA3 group and the HFD group was not statistically significant, suggesting that TA3 induces the recovery of serum lipid profiles in obese mice induced by an HFD. 

### 2.5. Timosaponin A3 Attenuates Hepatic Steatosis 

In accordance with several studies, in rodent models of non-alcoholic fatty liver disease, the consumption of an HFD is known to result in significantly brighter liver coloration [22]. To assess the effects of TA3 on hepatic tissue morphology, we observed tissue samples using visual analysis. As depicted in Figure 6A, liver tissues in the HFD group exhibited a lighter coloration compared to liver tissues in the ND group. Oral administration of TA3 significantly inhibited the morphological changes observed in liver tissues compared to the HFD group. Furthermore, liver weight increased in the HFD group compared to that in the ND group, while TA3 treatment led to a significant reduction in liver weight (Figure 6B). Moreover, in the HFD group, lipid droplets within hepatic cells appeared as small vacuoles, while in the TA3 group, the number and size of lipid droplets decreased compared to those in the HFD group (Figure 6C).

### 2.6. Timosaponin A3 Inhibits the White Adipose Tissue Generation and Size 

WAT mass is influenced by adipogenesis, a fundamental process in which preadipogenic cells differentiate into mature adipocytes [23]. The total masses of eWAT and iWAT in the HFD group were significantly higher (59.174 ± 6.186 and 47.571 ± 5.705 mg/kg, respectively) compared to those in the ND group (14.672 ± 1.049 and 9.716 ± 1.514 mg/kg). However, TA3 treatment significantly reduced the masses of eWAT and iWAT (46.528 ± 5.073 and 30.469 ± 6.582 mg/kg). Visual examination confirmed these differences (Figure 7A,B). To histologically analyze the effects of TA3 on lipid accumulation induced by HFD in WAT, H&E staining was performed. As a result, the size of adipocytes in subcutaneous and inguinal WAT significantly increased in the HFD group compared to those in the ND group, while TA3 treatment significantly reduced adipocyte size (Figure 7C,D). In summary, TA3 effectively reduces WAT mass and inhibits adipocyte hypertrophy induced by an HFD, demonstrating its potential as an anti-obesity agent. To elucidate the molecular mechanisms underlying the inhibitory effect of TA3 on adipogenesis, we examined the expression levels of key adipogenic proteins. The increased levels of FAS and SREBP1c induced by HFD were significantly reduced upon TA3 treatment (Figure 7E). These findings suggest that TA3 attenuates adipogenesis in WAT, thereby inhibiting the enlargement of adipocytes induced by HFD.

## 3. Discussion

Glucagon-like peptide-1 receptor agonists (GLP-1 RA) are incretin mimetics that, when blood sugar levels rise, induce pancreatic β cells to release insulin. They can regulate blood sugar levels and address conditions such as type 2 diabetes and obesity [24]. Liraglutide and Semaglutide are both GLP-1 RA medications used for the treatment of type 2 diabetes. The Danish pharmaceutical company Novo Nordisk developed Liraglutide, marketed as ‘Victoza,’ for daily subcutaneous injection as a treatment for type 2 diabetes [25]. They also made high-dose Liraglutide injection as ‘Saxenda’ for obesity treatment. In 2014, the U.S. Food and Drug Administration (FDA) granted approval for Saxenda as the first GLP-1 RA medication that can be used in conjunction with a low-calorie diet and regular exercise for obesity treatment in adults. In 2020, the approval was extended to include pediatric and adolescent populations for Saxenda [26]. 

In 2012, Novo Nordisk also developed Semaglutide, marketed as Ozempic. It is administered through a weekly injection and helps control blood sugar levels in diabetic patients. The FDA approved Ozempic as a diabetes drug in 2017 [27]. Over time, word spread among diabetic patients who were using Ozempic that they experienced a decrease in appetite and subsequent weight loss. This led to non-diabetic individuals becoming interested in using the injection as a potential weight loss aid. In response to Ozempic’s popularity for weight loss, Novo Nordisk made a high-dose Semaglutide injection particularly for weight management and released it under a new brand name, ‘Wegovy’, in 2021 [28]. 

Despite these high expectations and demands for its therapeutic effects, many studies found that the use of GLP-1 RA was associated with an increased risk of thyroid cancer [8], pancreatitis [9,10], gastroparesis, bowel obstruction [10], or biliary diseases [11]. Thus, efforts have been made to find natural compounds with similar actions but without the unwanted side effects of GLP-1 RA.

Traditionally, Jimo (*Anemarrhena asphodeloides* Bunge) has been used as one of the representative medical herbs to treat diabetic symptoms in Korean medicine. There is also scientific evidence that Jimo extract stimulates GLP-1 secretion in NCI-H716 cells [17]. GLP-1 enhances insulin secretion from pancreatic beta cells while simultaneously reducing glucagon levels. Beyond its effects on insulin and glucagon, it exhibits a wide range of actions, including the promotion of insulin sensitivity in adipose tissue, stimulation of energy expenditure and lipolysis in the adipose tissue, reduction of hepatic steatosis and liver lipid content, deceleration of gastric emptying and gastrointestinal motility, and augmentation of satiety while reducing appetite and food intake through actions in the brain [1]. Clinically, all these actions can be expected to induce weight loss when Jimo is prescribed in herbal formulas of Korean medicine. 

Timosaponin A3 (TA3), a natural steroidal saponin derived from Jimo, has exhibited various pharmacological activities in previous studies, including anti-inflammatory and anti-cancer effects [29]. As the major effective natural compound of Jimo, TA3 could be more effective than Jimo and safer than GLP-1 RA. Therefore, we assumed that TA3 could have both anti-diabetic and anti-obesity effects like GLP-1 RA. We wanted to confirm the action of TA3 to stimulate GLP-1 secretion and explore its role in obesity and related molecular mechanisms. To our knowledge, this is the first study to evaluate the potential anti-diabetic and anti-obesity effects of TA3 and the underlying mechanisms involved in both NCI-H716 cells and 3T3-L1 adipocytes. Moreover, we investigated the impact of TA3 on body weight and blood glucose levels in obesity induced by an HFD in mice.

To address safety concerns at an early stage of the investigation, the cytotoxicity of TA3 in NCI-H716 cells was evaluated centered on a concentration range up to 40 μM. Our choice of NCI-H716 cells for this study was based on their relevance as enteroendocrine L cells, which play a critical role in the secretion of GLP-1, a hormone known for its role in regulating glucose homeostasis and appetite [30]. Our results indicated that TA3 did not exert significant cytotoxic effects on NCI-H716 cells at concentrations up to 20 μM. Subsequently, we explored the potential of TA3 to stimulate GLP-1 secretion in NCI-H716 cells. GLP-1 is recognized for its anti-obesity and anti-diabetic effects, making it an attractive target for pharmacological interventions [3]. As a result, we observed a concentration-dependent increase in GLP-1 secretion upon TA3 treatment, further supporting its potential as a therapeutic agent for metabolic disorders. This suggests that TA3 may enhance GLP-1 secretion, which can contribute to improved glucose homeostasis and reduced appetite. To gain insight into the molecular mechanisms underlying TA3-induced GLP-1 secretion, we investigated key signaling pathways involved in GLP-1 regulation. Notably, we observed increased phosphorylation of PKAc and AMPK in response to TA3 treatment. AMPK is a well-known cellular energy sensor [31]. When activated, AMPK promotes energy balance by inhibiting catabolic pathways that consume adenosine triphosphate (ATP) while facilitating pathways that generate ATP [32]. In our study, we observed an increase in AMPK phosphorylation in NCI-H716 cells treated with TA3. This suggests that TA3 may induce a cellular energy-sensing response, potentially because of changes in the intracellular energy status. PKAc, also known as cyclic AMP (cAMP)-dependent protein kinase, is a key component of the cAMP signaling pathway. Activation of PKAc is typically mediated by an increase in the intracellular cAMP levels. PKAc plays a role in regulating numerous cellular processes, including hormone secretion [33]. In our study, we observed an increase in PKAc phosphorylation in response to TA3 treatment. This indicates that TA3 may influence the cAMP signaling pathway, potentially by modulating the cAMP levels or cAMP production in NCI-H716 cells. These findings suggest that TA3 may modulate GLP-1 secretion through the activation of AMPK and PKAc, leading to metabolic effects such as enhanced glucose uptake and improved insulin sensitivity.

Moving on to our investigation in 3T3-L1 adipocytes, we first assessed the cytotoxicity of TA3. Our results revealed no significant cytotoxic effects of TA3 at concentrations up to 10 μM over a 48-h exposure period, confirming its safety for further investigations. We then explored the impact of TA3 on adipogenesis, a critical process in the development of obesity. Adipocyte differentiation and lipid accumulation are central factors in the expansion of adipose tissue and subsequent weight gain. Using Oil Red O staining, we observed that TA3 treatment led to a concentration-dependent reduction in lipid droplet formation and absorbance, indicating its potential to inhibit adipogenesis in 3T3-L1 adipocytes. To gain deeper insights into the mechanisms and anti-lipogenic effects of TA3, we examined the expression of key lipogenic factors, including FAS and FABP4. Our results revealed that TA3 treatment resulted in reduced expression levels of these factors. FAS and FABP4 play crucial roles in lipid synthesis and transport within adipocytes, and their downregulation by TA3 suggests its ability to interfere with these processes [34,35]. Additionally, we investigated the impact of TA3 on transcription factors involved in adipocyte differentiation, namely C/EBPα and PPARγ. They are key regulators of adipogenesis and are associated with the formation of mature adipocytes [36]. Differentiation inducers typically increase the expression of these transcription factors, but TA3 treatment led to their downregulation. The suppression of their expression by TA3 further supports its role in inhibiting adipogenesis. Moreover, we assessed the expression of AMPK, a key regulator of adipocyte differentiation and energy metabolism. TA3 treatment resulted in a dose-dependent increase in AMPK expression. AMPK plays a vital role in cellular energy homeostasis and is known to inhibit adipogenesis [32]. The upregulation of AMPK by TA3 suggests its involvement in the suppression of adipocyte differentiation. Consequently, our findings confirm that TA3 stimulates GLP-1 secretion while also inhibiting lipogenesis and adipogenesis in vitro.

Then, we conducted a study using a mice model fed an HFD. Our findings demonstrated that the administration of TA3 significantly mitigated the substantial weight gain observed in mice subjected to an HFD, ultimately resulting in a notable difference by week 12. This effect was accompanied by noticeable changes in the animals’ body images, suggesting a visible reduction in adiposity due to TA3 treatment. Interestingly, while the reduction in food consumption in the TA3-treated group did not reach statistical significance when measured in grams, when considering calorie intake (kcal), the TA3-treated group exhibited a substantial reduction compared to that in the HFD group. This implies that TA3 might influence the quantity and quality of food intake or energy utilization, contributing to its anti-obesity effects.

Our research results provide insights into the potential therapeutic effects of TA3 in addressing hyperlipidemia and related metabolic issues in an obesity-induced mouse model. To assess actual glucose metabolism, an OGTT was conducted, and AUC was calculated for comparison among the groups based on the OGTT results. As a result, the OGTT revealed that the HFD group exhibited prolonged elevation in blood glucose concentrations following glucose administration compared to that in the ND group, which is indicative of impaired glucose metabolism associated with an HFD. However, the TA3 group demonstrated a significant reduction in blood glucose levels compared to that in the HFD group. When comparing AUC, the TA3 group showed a significant decrease compared to that in the HFD group, indicating that TA3 may have a beneficial impact on glucose regulation, potentially alleviating high blood sugar associated with a high-fat diet. Furthermore, our findings indicate that TA3 has a profound effect on serum lipid profiles. We observed a significant elevation in all lipid profile factors, including AST, ALT, TC, TG, and LDL, in the HFD-treated group compared to the corresponding factors in the ND group. However, the TA3 group exhibited a remarkable reduction in these parameters, indicating its potential to ameliorate hyperlipidemia. This overall improvement in the lipid profile suggests that TA3 may effectively counteract the adverse effects of an HFD on lipid metabolism.

Furthermore, our investigation distinctly revealed TA3’s protective effects against hepatic steatosis, as was evident from the visual assessment of liver tissues. Hepatic steatosis, often induced by an HFD, is characterized by the excessive accumulation of fat within liver cells. One prominent hallmark of hepatic steatosis is the alteration in liver tissue coloration, typically resulting in a brighter appearance [22]. In our study, we observed that liver tissues in the HFD group exhibited this characteristic light coloration, indicating hepatic lipid accumulation. However, administration of TA3 significantly mitigated these morphological changes in liver tissues, suggesting its potential to ameliorate hepatic steatosis. Moreover, the liver weight noticeably increased in the HFD group compared to that in the ND group. Nevertheless, TA3 administration led to a significant reduction in liver weight, signifying its potential to counteract HFD-associated hepatomegaly. Further, we observed that in the HFD group, lipid droplets within hepatic cells appeared as small vacuoles, indicative of fat accumulation. In contrast, the TA3 group exhibited a reduction in both the number and size of lipid droplets compared to those in the HFD group, although the size difference was not substantial. These results suggest that TA3 may alleviate hepatic lipid droplet accumulation, a characteristic feature of simple steatosis.

Moreover, our investigation into the effects of TA3 on WAT mass and adipocyte size yielded significant findings. TA3 effectively reduced the mass of both inguinal and epididymal WAT. This outcome carries substantial clinical relevance, given the close association between reduced adipose tissue mass and improvements in insulin sensitivity and dyslipidemia, both of which are paramount in addressing obesity-related comorbidities. The reduction in adipose tissue mass, particularly visceral fat (eWAT), is of particular importance owing to its implications for metabolic health. eWAT is metabolically active and plays a central role in the development of insulin resistance and adverse lipid profiles [37]. Therefore, interventions that can effectively reduce eWAT mass have the potential to ameliorate obesity-related metabolic disturbances. Histological analysis further revealed that TA3 treatment significantly reduced the size of adipocytes in both eWAT and iWAT. The hypertrophy of adipocytes is a hallmark of obesity and is linked to various metabolic complications. The reduction in adipocyte size observed in our study suggests that TA3 has the capacity to counteract adipocyte hypertrophy induced by an HFD. To gain insights into the molecular mechanisms underlying the anti-adipogenic effects of TA3, we examined the expression levels of key adipogenic proteins. Our results demonstrated that TA3 significantly reduced the expression of FAS and SREBP1c, both of which are well-known regulators involved in fat synthesis. The reduction in the expression of these adipogenic factors aligns with the observed inhibition of fat synthesis and lipid accumulation in the adipose tissue.

## 4. Materials and Methods

### 4.1. Reagents 

TA3 was purchased from Wuhan Chem Faces Biochemical Co., Ltd. (Wuhan, China). The lysis buffer and cell fractionation kit were purchased from Cell Signaling Technology (Beverly, MA, USA). Dexamethasone (DEX), insulin, and methylisobutylxanthine (IBMX) were purchased from Sigma-Aldrich (St. Louis, MO, USA). Enhanced chemiluminescence (ECL) solution was obtained from DOGEN (Seoul, Republic of Korea). Antibodies for SREBP1c (sc-13551), β-actin (sc-47778), and secondary antibodies (sc-516102 and sc-2004) were purchased from Santa Cruz Biotechnology (Santa Cruz, CA, USA) and the other antibodies were purchased from Cell Signaling Technology. These antibodies were as follows: fatty acid synthase (FAS) (3180s), fatty acid-binding protein 4 (FABP4) (2120s), C/EBPα (8187s), peroxisome proliferator-activated receptor gamma (PPARγ) (2435s), AMP-activated protein kinase (AMPK) (2532s), phospho-AMPK (2535s), protein kinase A catalytic subunit (PKAc) (4782s), and phospho-PKAc (p-PKA C) (4781s).

### 4.2. Cell Culture 

Two cell lines, NCI-H716 and 3T3-L1, were utilized in this study. NCI-H716 and 3T3-L1 cells were procured from the Korean Cell Line Bank and cultured in a 5% CO_2_, 37 °C cell incubator. The NCI-H716 cell line is derived from a 33-year-old male with cecal adenocarcinoma, and when cultured with a specific extracellular matrix, these cells undergo endocrine differentiation and express neuroendocrine markers like chromogranin A. This cell line serve as a valuable model for researching the regulation of GLP-1 secretion [38]. And 3T3-L1 cell line primarily originates from embryonic fibroblasts of mice. It has the ability to transform into adipocytes through differentiation processes, making it widely used in the study of cell differentiation, adipogenesis, metabolism of fat cells, and related endocrine functions [39]. The culture medium was supplemented with 10% fetal bovine serum (FBS) and 100 U/mL penicillin.

NCI-H716 cells were cultured in RPMI medium, maintaining them in a floating state. At 3 days before the experiments, NCI-H716 cells were seeded in 12-well plates pre-coated with matrigel, using Dulbecco’s Modified Eagle Medium (DMEM) as the culture medium. These cells were allowed to differentiate for 2 days and were subsequently subjected to 16 h of serum starvation. Then, TA3, prepared in 1 mM CaCl_2_-containing phosphate-buffered saline (PBS) according to the experimental conditions, was added to the cells and incubated for 2 h. The culture medium was employed to assess the GLP-1 levels, while cells were utilized for protein-level analysis.

Moreover, 3T3-L1 cells were cultured in DMEM medium. Cells were seeded in 6-well plates at a density of 8 × 10^4^ cells/well and medium changes were performed until the wells were fully occupied. For differentiation, cells were cultured in differentiation medium (0.5 mM IBMX, 0.5 μM DEX, and 10 μg/mL insulin, abbreviated as MDI). Subsequently, the medium was changed every 2 days with maintenance medium containing 10 μg/mL insulin and TA3 (0–10 μM) until the 8th day of culture.

### 4.3. 3-(4,5-Dimethylthiazol-2-yl)-2,5-Diphenyltetrazolium Bromide (MTT) Assay

Cellular toxicity of TA3 was assessed using the MTT solution in both NCI-H176 and 3T3-L1 cells. This experiment was conducted with reference to previous research [40]. Briefly, cells were seeded in 96-well plates and allowed to culture overnight. Subsequently, the culture medium was replaced with media containing various concentrations of TA3 (0, 2.5, 5, 10, 20, and 40 μM). Following incubation under the specified experimental conditions, MTT solution (2 mg/mL) was added to achieve a final concentration of 0.5 mg/mL, and the cells were further incubated for 2 h. Formazan crystals that formed were dissolved in DMSO, and absorbance was measured at 540 nm using a microplate reader.

### 4.4. ELISA (Enzyme-Linked Immunosorbent Assay)

For the analysis of the GLP-1 and insulin levels, ELISA kits were employed, following the protocols provided by the respective manufacturers.

### 4.5. Western Blot

Protein extraction from cells was carried out using cell signaling lysis buffer, while tissues were processed with RIPA buffer. This experiment was conducted with reference to previous research [41]. Primary antibodies, diluted at a 1:1000 ratio, were applied for overnight incubation at 4 °C. Then, the membrane underwent washing with TBS-T and was subsequently incubated with secondary antibodies, diluted at a 1:10,000 ratio, for 1 h. The band was visualized with ECL solution using an ImageQuantTM LAS500 chemiluminescence (GE Healthcare Bio-Sciences, Chicago, IL, USA) and quantified using Image J software (https://imagej.net/ij/, US National Institute of Health, Bethesda, MD, USA). The primary antibodies used included C/EBPα, PPARγ, SREBP1c, FAS, p-AMPK, AMPK, p-PKAc, PKAc, FABP4, and β-actin.

### 4.6. Oil Red O Assay

In accordance with the experimental conditions, 3T3-L1 cells treated with TA3 were washed with PBS. Fixation was carried out by immersing the cells in 10% formalin for a duration of 1 h. Then, the cells were washed with 60% isopropanol and allowed to air dry completely. A working solution of 60% Oil Red O, dissolved in distilled water (DW), was applied to the cells for a period of 30 min. After staining, the cells were rinsed three times with DW. Images of lipid droplet-stained cells were taken using an Olympus IX71 microscope (Olympus, Tokyo, Japan). To quantify lipid accumulation, the stained cells were dissolved in 100% isopropanol, and absorbance was measured at 490 nm using a microplate reader

### 4.7. Animals

All animal-related procedures were conducted following the review and approval of the Institutional Animal Care and Use Committee of Kyung Hee University (Approval No. KHSASP-23-308). Five-week-old male C57BL/6J mice were procured from Daehan BioLink (DBL, Chungcheongbuk-do, Republic of Korea). The animals were provided access to sterilized mouse chow and water ad libitum. They were acclimated in a controlled environment for 1 week before the commencement of experiments. During this period, the mice were maintained in a room with a consistent 12-h light–dark cycle, controlled temperature, and humidity.

### 4.8. In Vivo Experiments

Following an adaptation period, mice were provided with a high-fat chow diet (D12492, RD 60% fat calorie) for 12 weeks, except for the normal-diet (ND) group. Starting from the 9th week of the high-fat chow diet, TA3 was orally administered at a dose of 10 mg/kg three times a week, with PBS serving as the vehicle. Throughout the study, all mice underwent regular weight measurements three times a week, while food intake was assessed once a week. At 1 week before the conclusion of the experiment, an Oral Glucose Tolerance Test (OGTT) was conducted to evaluate blood glucose levels. On the final day of the experiment, all mice were euthanized, and serum, liver tissue, epididymal white adipose tissue (eWAT), and inguinal white adipose tissue (iWAT) were collected. The weight of each tissue was recorded for further analysis.

### 4.9. OGTT

To conduct the OGTT at a dosage of 5 g/kg, mice were subjected to a 16-h fasting period. Glucose (5 g/kg) was administered orally, and blood glucose levels were measured at six different time points via tail vein. Measurements were taken using the Accu-Chek Performa system (Roche Diagnostics, Basel, Switzerland) at the following time intervals: 0 (prior to oral glucose administration) and 15, 30, 60, and 120 min after oral glucose administration. The blood glucose levels were recorded at least twice at each time point to ensure accuracy and consistency in the data. To evaluate the OGTT, the areas under the curves (AUCs) were calculated from 0 min to 120 min.

### 4.10. Serum Analysis

Blood samples were collected in 1.5 mL microtubes pre-coated with EDTA before dissection. Following collection, the samples were centrifuged at 2000 rpm for 10 min at 4 °C to isolate the serum, which was subsequently stored at −80 °C. To analyze total cholesterol (TC), triglycerides (TG), low-density lipoprotein (LDL), and high-density lipoprotein (HDL), external laboratory services provided by DKKOREA (Seoul, Republic of Korea) were employed.

### 4.11. H&E (Hematoxylin and Eosin) Staining

The collected eWAT, iWAT, and liver tissues were fixed in 10% formalin. After fixation, the tissues were rinsed in running tap water for 24 h and subsequently paraffinized. Paraffin blocks were prepared by embedding, and 4-μm sections were cut. The sections were deparaffinized in xylene, rehydrated through 100%, 90%, 80%, and 70% EtOH, and washed in PBS. Subsequently, the sections were stained with Harris’ hematoxylin for 2 min and Eosin Y solution for 30 s. Dehydration was carried out using 70%, 80%, 90%, and 100% EtOH. Then, the sections were cleared in xylene to remove EtOH, and the dehydrated sections were mounted using DPX Mountant (Sigma). Images of the sections were captured using an Olympus IX71 microscope (Olympus).

### 4.12. Statistical Analysis

The significance of each comparison was analyzed by an unpaired t-test (two-tailed) using GraphPad Prism software (Version 5.0, San Diego, CA, USA). All experimental data are expressed as means ± standard deviations.

## 5. Conclusions

In conclusion, our study provides comprehensive evidence for the potential anti-obesity and anti-diabetic properties of TA3. It exerts its effects by promoting GLP-1 secretion, inhibiting adipogenesis, reducing body weight gain, improving glucose homeostasis, modulating lipid profiles, and attenuating hepatic steatosis. These findings suggest that TA3 holds promise as a therapeutic agent for the treatment and prevention of obesity and its associated comorbidities.

## Figures and Tables

**Figure 1 ijms-25-02914-f001:**
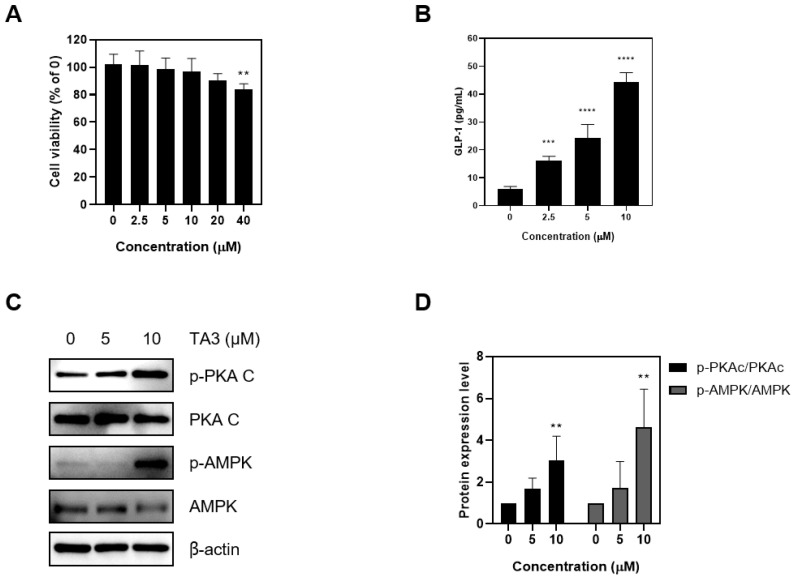
Effects of Timosaponin A3 on NCI-H716 cell lines. (**A**) Cell proliferation was assessed in NCI-H716 cell lines using MTT analysis following treatment with TA3. (**B**) The GLP-1 levels were evaluated in NCI-H716 cell lines after treatment with TA3 under experimental conditions by removing the supernatant and performing ELISA analysis. (**C**) The impact of TA3 treatment on the phosphorylation of PKAc and AMPK, which are upstream signals for GLP-1 secretion, was examined. (**D**) Protein expression levels. Data are presented as means ± SEM. Significance was determined by comparing the data with those in the control group: ** *p* < 0.01, *** *p* < 0.001, and **** *p* < 0.0001.

**Figure 2 ijms-25-02914-f002:**
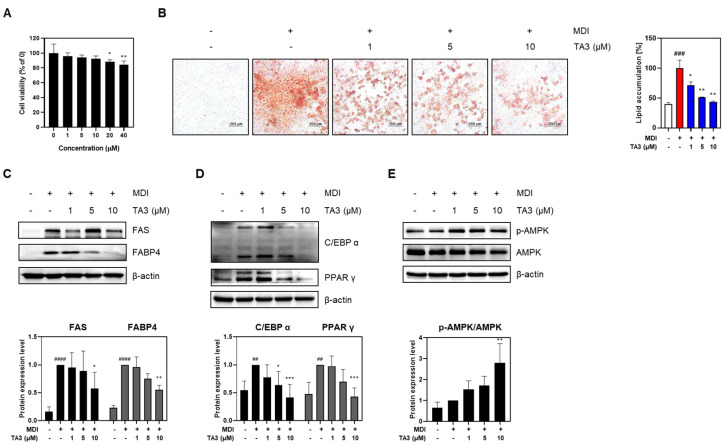
Effects of Timosaponin A3 on 3T3-L1 adipocytes. (**A**) Cell viability in 3T3-L1 adipocytes was evaluated using MTT analysis following treatment with TA3. (**B**) Following treatment under experimental conditions, 3T3-L1 adipocytes were stained with Oil red O dye, and lipid droplets were imaged at 100× magnification under a microscope. The absorbance values were measured after dissolution in 100% isopropanol. (**C**) Expression levels of factors related to lipogenesis, (**D**) factors related to adipogenesis, and (**E**) protein expression levels of the AMPK signal. Data are presented as means ± SEM. Significance was determined by comparing with the control group: ## *p* < 0.01, ### *p* < 0.05 and #### *p* < 0.0001. Additionally, significance was determined by comparing the data with those in the MDI group: * *p* < 0.05, ** *p* < 0.01 and *** *p* < 0.001.

**Figure 3 ijms-25-02914-f003:**
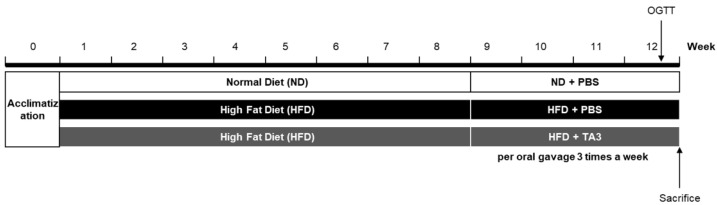
In vivo experiment scheme.

**Figure 4 ijms-25-02914-f004:**
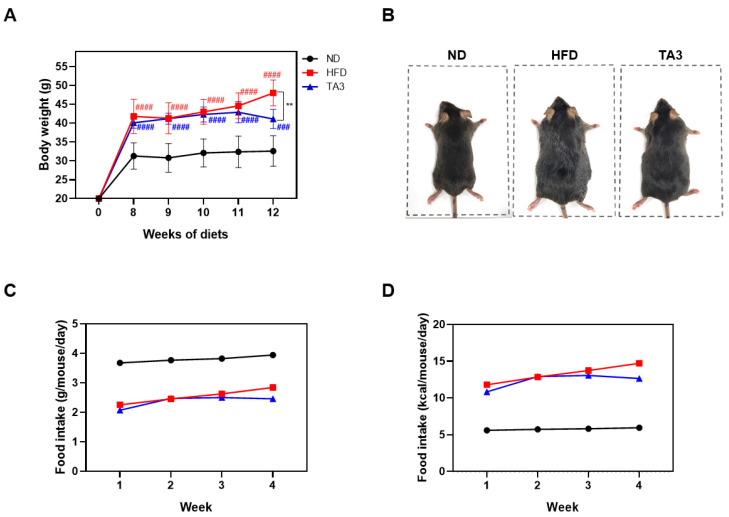
Effects of Timosaponin A3 administration on body weight and food intake profile. (**A**) Comparison of body weight in each group from the start of drug treatment at Week 0 (0 W) to Week 12 (12 W). (**B**) Macroscopic comparison of body image in each group. (**C**) Food intake per gram (g) of body weight. (**D**) Food intake per kilocalorie (kcal). Data are presented as means ± SEM. Significance was determined by comparing with the ND group: ### *p* < 0.001 and #### *p* < 0.0001. Additionally, significance was determined by comparing the data with those in the HFD group: ** *p* < 0.01.

**Figure 5 ijms-25-02914-f005:**
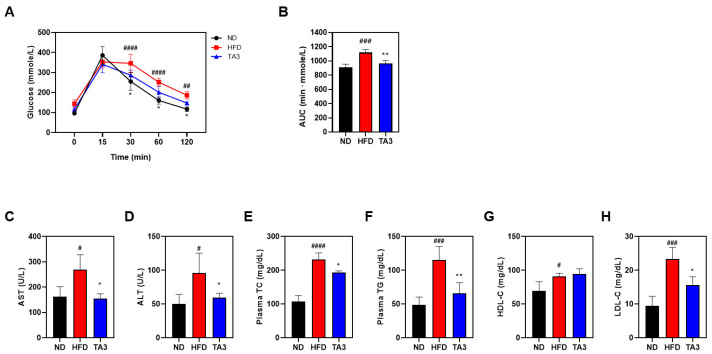
Effects of Timosaponin A3 on glucose homeostasis and serum lipid profile of high-fat diet mice model. (**A**) Changes in OGTT following treatment with Vehicle and TA3 in HFD-induced C57BL/6 mice. (**B**) AUC 0–120 min values of OGTT were calculated. (**C**) AST, (**D**) ALT, (**E**) TC, (**F**) TG, (**G**) HDL, and (**H**) LDL levels were measured in the serum of all mice in the experimental groups. Data are presented as mean ± SEM. Significance was determined by comparing the data with those of the ND group: # *p* < 0.05, ## *p* < 0.01, ### *p* < 0.001, and #### *p* < 0.0001. Additionally, significance was determined by comparison with the HFD group: * *p* < 0.05 and ** *p* < 0.01.

**Figure 6 ijms-25-02914-f006:**
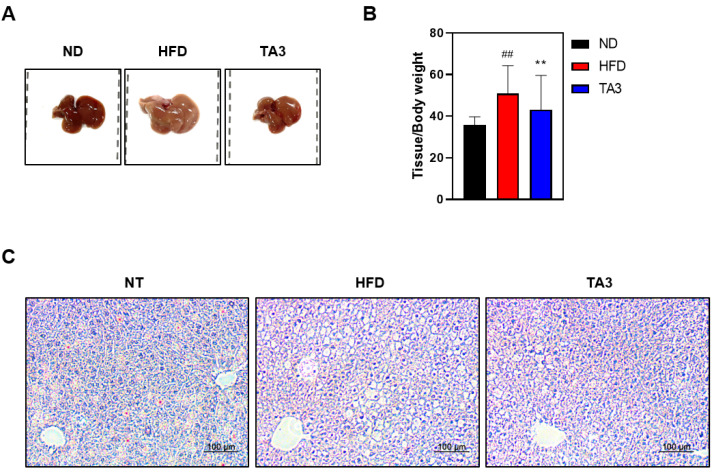
Inhibitory effect of lipid droplet accumulation in liver tissue by Timosaponin A3. (**A**) Macroscopic analysis of mouse liver tissue. (**B**) Liver weight-to-body weight ratio. (**C**) Representative liver tissues from mice in each group were fixed, paraffin-embedded, and stained with H&E solution. Subsequently, they were imaged at a 200× magnification under a microscope. Data are presented as means ± SEM. Significance was determined by comparing the data with those of the ND group: ## *p* < 0.01. Additionally, significance was determined by comparing data with those of the HFD group: ** *p* < 0.01.

**Figure 7 ijms-25-02914-f007:**
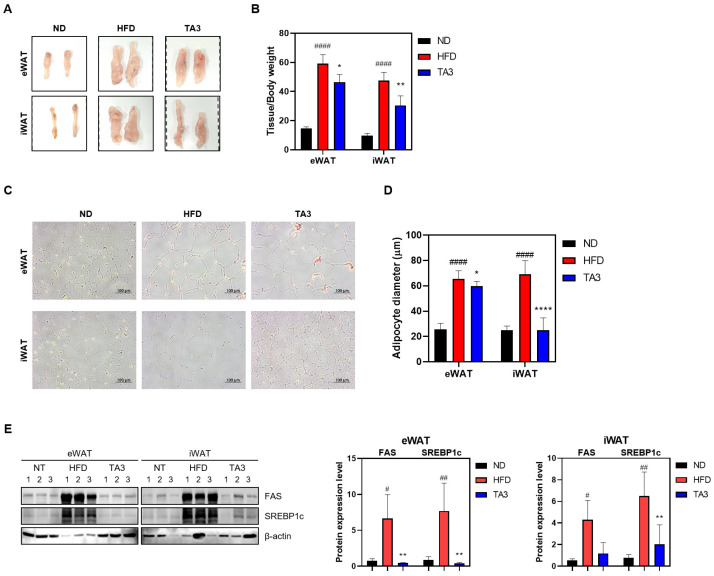
Effects of TA3 on white adipose tissue in high-fat diet-induced obesity mice model. (**A**) Macroscopic analysis of eWAT and iWAT. (**B**) Weight of each adipose tissue relative to body weight. (**C**) Mouse adipose tissue from each group was fixed, paraffin-embedded, and stained with H&E solution. Subsequently, they were imaged at 200× magnification under a microscope, and adipocyte diameter was quantified (**D**). (**E**) Expression of lipogenesis-related factors within adipose tissue. Data for each group are presented as means ± SEM. Significance was determined by comparing the data with those in the ND group: # *p* < 0.05, ## *p* < 0.01, and #### *p* < 0.0001. Additionally, significance was determined by comparing the data with those in the HFD group: * *p* < 0.05, ** *p* < 0.01, and **** *p* < 0.0001.

## Data Availability

All data presented in this study are available from the corresponding author upon reasonable request.

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
