# Peer review of "Timosaponin A3 Induces Anti-Obesity and Anti-Diabetic Effects In Vitro and In Vivo"

_ijms, 2024, doi:10.3390/ijms25052914_

Round 1
Reviewer 1 Report
Comments and Suggestions for Authors
The manuscript “Timosaponin A3 induces anti-obesity and anti-diabetic effects 2 in vitro and in vivo” explore the possibility for Timosaponin A3 of being used as a natural compound for tackling obesity, diabetes, and associated metabolic disorders.
I consider relevant for the fields the effects of Timosaponin A3 on diabetes and obesity since until now are largely unexplored. Results obtained by authors are relevant because they have identified a natural molecule that has similar effects to GLP-1 RA which despite high expectations and demands for its therapeutic effects, has numerous side effects. The authors demonstrated that TA3 may be considered a potential anti-obesity agent providing a novel approach for therapeutic intervention. In my opinion , all main questions posed were addressed and by which specific experiments.
In my opinion, this is a well-conducted study using both in vitro and in vivo models.
However minor revisions are requested:
1) In Figure 2 Panel D, the Western image of the C/EBP a protein is unclear. The authors should provide an image that better highlights the protein band
2) In Figure 2 the SD value of the control group does not seem to justify the significance reported for 20 and 40 mM concentrations. The authors should justify these results
Authors conclude that their findings suggest that TA3 holds promise as a therapeutic agent for the treatment and prevention of obesity and its associated comorbidities.
Author Response
Reviewer #1: The manuscript “Timosaponin A3 induces anti-obesity and anti-diabetic effects 2 in vitro and in vivo” explore the possibility for Timosaponin A3 of being used as a natural compound for tackling obesity, diabetes, and associated metabolic disorders.
Point 1: In Figure 2 Panel D, the Western image of the C/EBP a protein is unclear. The authors should provide an image that better highlights the protein band.
(Response) Following your advice, I have replaced it with a clearer band.
Point 2: In Figure 2 the SD value of the control group does not seem to justify the significance reported for 20 and 40 mM concentrations. The authors should justify these results
(Response) The results in question were analyzed using the statistical software GraphPad Prism, employing a One-way ANOVA followed by Tukey's multiple comparisons test. Upon reviewing the raw statistical data, significant differences were observed when comparing the control group (0 µM) to the groups treated with 20 and 40 µM concentrations, justifying the significance reported.

Reviewer 2 Report
Comments and Suggestions for Authors
Dear author,
thank you for submitting your work, „Timosaponin A3 induces anti-obesity and anti-diabetic effects 2
in vitro and in vivo“.
Here are some comments for improvement:
Please align space in text between the lines, it varies throughout the text.
Please do not start a sentence with an abbreviation, check lines: 52; 170, 190, 247, 270, 284, 300, 309,
Please define the cells and than put the abbreviation, line 54.
Not sure what „patogenic heat“ is, please explain. Line 61.
Put space between the words, line 66.
Define on which cells cytotoxicity was tested, please, Line 80.
Define abbreviations before use, Line 86, 111, 113, 134, 149, 154, 159, 190, 202, 299, 303,
Define cells with cell line code, Line 93. Change the font size of the picture explanation.
Explain the desing of the experiment, why is it in terms of time, done in 8 days and than 48h? Line 103.
Write in vivo in italic, line 132.
Do not put abbreaviation in the title, line 170
Lines 213 – 230 should be in Introduction part, lines 241- 244 as well,
There is no starting part regarding the figure 7 in dsicussion, please add.
Reference missing, line 249 – 254. This seems like an introduction part as well.
Line 255 – 263 should be part of introduction, not discussion.
Write first a full name and than abbreviation of the test, line 402.
Methods should be writen in more concise way and with logic of the experiment, please revise.
Reagrds,
Comments on the Quality of English Language
Dear authors, English should be improved, more in a style, than just grammar.
Author Response
Reviewer #2: Thank you for submitting your work, “Timosaponin A3 induces anti-obesity and anti-diabetic effects 2 in vitro and in vivo”. Here are some comments for improvement:
Point 1: Please align space in text between the lines, it varies throughout the text.
(Response) Following your advice, I have unified and aligned the text spacing between lines throughout the manuscript.
Point 2: Please do not start a sentence with an abbreviation, check lines: 52; 170, 190, 247, 270, 284, 300, 309,
(Response) TA3 is an abbreviation for Timosaponin A3, which was introduced earlier in the text. In response to your comment, I have revised line 170. To ensure consistency throughout the document, given that TA3 is used in various sections, I have changed all instances of TA3 in the subheadings to its full name, Timosaponin A3, to maintain uniformity.
Point 3: Please define the cells and than put the abbreviation, line 54.
(Response) NCI-H716 is not an abbreviation. To provide clarity on the origin of these cells and to maintain consistency, I have also included the origin of the 3T3-L1 cell line in the Materials and Methods section (lines 386-393).
Point 4: Not sure what „patogenic heat“ is, please explain. Line 61.
(Response) Following your advice, I have added further explanation to the content in question.
Point 5: Put space between the words, line 66.
(Response) Following your advice, I have revised that section in the manuscript.
Point 6: Define on which cells cytotoxicity was tested, please, Line 80.
(Response) Following your advice, I have revised that section in the manuscript.
Point 7: Define abbreviations before use, Line 86, 111, 113, 134, 149, 154, 159, 190, 202, 299, 303,
(Response) The abbreviations mentioned (FAS, FABP4, C/EBP alpha, PKAc, PPAR gamma, AMPK) are all defined in the Abstract and within the Reagents section of the Materials and Methods. Furthermore, the abbreviations ND, HFD, and TA3 are defined in the Animals subsection of the Materials and Methods.
Point 8: Define cells with cell line code, Line 93. Change the font size of the picture explanation.
(Response) Following your advice, I have revised that section in the manuscript.
Point 9: Explain the desing of the experiment, why is it in terms of time, done in 8 days and than 48h? Line 103.
(Response) The design of the experiment, specifically the choice to conduct the MTT assay for 48 hours, was based on the necessity to change the media every 48 hours. This requirement informed our decision to structure the experiment in this manner. Additionally, the overall schedule, including the 8-day duration, was determined by the need to allow for the adipocytes to undergo differentiation. This process necessitates a cultivation period of up to 8 days to achieve the desired outcomes.
Point 10: Write in vivo in italic, line 132.
(Response) Following your advice, I have revised that section in the manuscript.
Point 11: Do not put abbreaviation in the title, line 170
(Response) Following your advice, I have revised that section in the manuscript.
Point 12: Lines 213 – 230 should be in Introduction part, lines 241- 244 as well,
(Response) Originally located in the Introduction, the content pertaining to GLP-1 has been moved to the Discussion section in order to emphasize the information on GLP-1 more effectively.
Point 13: There is no starting part regarding the figure 7 in dsicussion, please add.
(Response) The content in question has been relocated to lines 352-369 in the Discussion section.
Point 14: Reference missing, line 249 – 254. This seems like an introduction part as well.
(Response) I have inserted the reference as suggested. Thank you for your detailed advice. However, due to the flow of content, it has been positioned in the Discussion section rather than the Introduction, as it aligns more coherently with the narrative and analysis presented there.
Point 15: Line 255 – 263 should be part of introduction, not discussion.
(Response) Thank you for your detailed advice. However, I also believe that, for the sake of content flow, this particular content is more appropriately placed in the Discussion section rather than the Introduction. This placement better suits the context and progression of the arguments and findings presented.
Point 16: Write first a full name and than abbreviation of the test, line 402.
(Response) Following your advice, I have revised that section in the manuscript.
Point 17: Methods should be writen in more concise way and with logic of the experiment, please revise.
(Response) Following your advice, I have revised that section in the manuscript.
